# Optimization of Mechanical Properties during Cold Rolling and Wrap-Forming for Metal Helical Tubes

**DOI:** 10.3390/ma17194832

**Published:** 2024-09-30

**Authors:** Shuyang Zhang, Yuming Chen, Yan Wang, Hui Yang, Congfa Zhang, Jintong Liu, Zongquan Deng

**Affiliations:** 1State Key Laboratory of Robotics and System, Harbin Institute of Technology, Harbin 150001, China; dengzq@hit.edu.cn; 2Beijing Institute of Spacecraft System Engineering, Beijing 100094, China; congfa_zh@163.com (C.Z.); jt_liu1@163.com (J.L.); 3School of Mechanical Engineering, Yanshan University, Qinhuangdao 066004, China; chenyumingzy@163.com

**Keywords:** in-orbit roll-forming strip, helical wrapping tube, torque, maximum stress, equivalent plastic strain, response surface method, optimization

## Abstract

The proposed method involves the utilization of a strip roll-forming technique to fulfill the specific requirements for constructing large-scale structures in orbit and space station trusses during extraterrestrial exploration. This involves progressively rolling out a metal strip with an L+V-shaped locking edge through multiple passes of forming rolls featuring different section shapes. The process of helical locking enables the formation of a slender, spiral-shaped tube, which can be utilized for the in-orbit assembly of exceptionally large structures. The proposed approach introduces a two-step optimization method to enhance the maximum stress, equivalent plastic strain, and wrapping torque of roll-forming L+V cross-sections and wrap-forming helical tubes. The optimized sample points in the second step are established based on the optimal roll distance and roll gap obtained in the first optimal step, resulting in the determination of the optimal folding height and V-shaped angle of the L+V strip cross-section after optimization. The design of the strip wrapping configuration that follows is highly dependent on these four parameters.

## 1. Introduction

The space truss, an integral component of spacecraft structures, comprised a specific number of one-dimensional rods interconnected in a predetermined direction within three-dimensional space. It has found extensive applications in the construction of deep space exploration bases and the augmentation of space station functionalities. The space truss structure utilized for in-orbit operation in current spacecraft typically ranges from 10 to 100 m [1,2,3]. Future space exploration will necessitate utilizing larger, lightweight, high-performance, and cost-effective space trusses, such as in-orbit service platforms. The completed structures are anticipated to possess a geometric scale ranging from 0.1 to 10 km [4,5,6]. The current methods for deploying and assembling in-orbit are significantly limited [7], thus necessitating the development of novel construction techniques. The current operational mode of ground preparation, rocket transportation, and space application poses significant challenges due to the limited volume and mass of the rocket, as well as the strict mechanical constraints during launch. The utilization of miniaturized manufacturing equipment for metal tube fitting in space, employing strip metal coils as raw materials, has the potential to facilitate the in-orbit wrapping of metal tube fittings, and subsequently provide a continuous supply of tube fitting units for the in-orbit assembly of super-sized structures. The helical lock is tightly wrapped around the tube body, enhancing the rigidity of tube fittings and improving their durability. The applications of this technology are widely utilized in various scenarios.

In forming tube fittings, the bending of metal strips induces elastoplastic deformation. Theoretical methods were utilized to solve thin-walled sheet displacement, stress, and strain under pure bending conditions [8]. Based on Brush et al.’s research on the theory of metal bending, Bai [9,10] developed a theoretical approach to investigate the bending behavior of composite materials, taking into consideration the geometric characteristics of their cross-sectional shape. The proposed method was validated through finite element (FE) modeling, and the obtained results exhibited excellent agreement with the theoretical approach. Safdarian [11] investigated the effect of some roll-forming parameters of channel section on the edge longitudinal strain and bow defect of products. The analytical model developed by Kilz [12] established a quantitative correlation between profile defects and the underlying longitudinal strain distributions. The elastic helical folding tube was investigated by Ben-Abu [13], who developed a theoretical model to describe the deformation of the helical elements. Yang [14] investigated the effects of roll gap and roll spacing on the forming of asymmetric cross-sections of strips using ABAQUS simulation and experiments. Liu [15] found that the deformation behavior of the double-wall brazed tube in the multi-pass roll-forming process had an important impact on the forming accuracy. The study conducted by Qazani [16] revealed that the energy utilization and required maximum torque of the forming line can be influenced by the effective indexes in the cold roll-forming procedure. Bidabadi [17] investigated how to reduce defects, such as profile bowing. The presence of a longitudinal bow, the cause of flange height deviation, spring-back, and the identification of thinning locations were investigated by Murugesan [18] in the cold roll-forming of symmetrical short U-profile sheets. Tajik [19] studied the twist defect by finite element (FE) analysis for asymmetrical channels with different flange lengths, and found that the twist defect was observed to result from a torsional torque. Weiss [20] studied the roll-forming of two coils exhibiting a coil set, and found significant differences in shape defects in the industrial roll-forming of a particular channel section depending on whether the strip was fed in with the convex side upwards or downwards. Spathopoulo [21] produced a desired formed shape blank and investigated the spring-back by FE analysis and artificial neural network approach.

This study presents the design of a compact and lightweight helical forming device for metal tubes in orbital applications. The structure comprises four sets of rollers, a wrapping mandrel, a locking roller, and a flat strip storage component. The rolling and wrapping process of the metal strip is optimized through a two-step approach. Optimizing the flat strip roll-forming V+L section is carried out, determining the optimal roll gap and roll space. The FE model of the strip helical wrapping is established by optimizing the roll gap and roll space in the first step, while simultaneously optimizing the height of the L-side and V-angle of the strip cross-section. The second step involves studying the optimized parameters of the sample points determined by the experimental of design, and analyzing the influence rules of the L-side height and V-angle on both wrapping torque and maximum stress.

## 2. Problem Description

The three-dimensional configuration of the metal strip helical apparatus forming is shown in Figure 1. The strip passes through the flat strip storage component, slot bending and rolling mechanism, wrapping, and locking device to form a helical tube in the direction of the mandrel. A schematic diagram of metal cold wrap- and roll-forming apparatus is shown in Figure 2. During the wrap- and roll-forming process, the strip is transferred from the strip storage component to the lock slot bending device and sequentially passes through four groups of rollers. These four sets of rollers compress the two edges of the flat strip out of the lock groove. The strip with a locking slot then goes through the tensioning device before entering into the wrapping locking device. With the combined action of a wrapping mandrel, wrapping die, and three different kinds of rollers, helical wrapping occurs for strips with locking edges, while simultaneously achieving edge locking and flattening. This continuous forming process results in the metal helical tube being produced.

The L+V section is formed after the strip passes through four sets of rollers, as illustrated in Figure 3. The L-shape right angle and the V-shape angle are interlocked during helical wrapping to achieve continuous locking, thereby forming a helical metal tube.

The configuration of metal strip helical forming mainly comprises two stages: roller cross-section formation and coil-locking synthesis of the helical tube. The two stages are optimized individually, and the flowchart of the optimization is showcased in Figure 4. The response surface method (RSM) is applied to the appropriate surrogate models. The selection of an appropriate experimental design method is made to obtain the necessary sample points. Then, the accuracy of the RS model is checked. If the accuracy is sufficiently high, then the RS models will be optimized through the utilization of a suitable optimization algorithm. Otherwise, a new RS surrogate model will be constructed, or additional design points will be added. If the accuracy of the optimal results is good enough, then the flowchart stops. If not, then a new RS surrogate model is selected, or additional design points are added.

## 3. Response Surface Method

The response surface method is employed to establish the surrogate models that describe the behavior of the spiral wrap- and roll-forming, aiming to optimize the parameters and minimize computational costs. The responses for the strips are maximum stress *S_r_*_-*max*_ and equivalent plastic strain *S_ep_* during the first roll-forming process. The responses for helical coiling torque *M_sc_* and maximum stress *S_c_*_-*max*_ during the second helical coil-forming process can be written in terms of a series of basic functions as
(1)$$\tilde{y}(x,y)=\sum_{i=1}^{S}\beta_{i}\varphi_{i}(x,y)+\varepsilon$$
where $\tilde{y}(l,\omega )$ represents the responses, *n* is the number of basic functions *φ_i_*(*x*, *y*), *i* is the number of the design variables, *β_i_* is the coefficient of the basic functions, and *ε* represents the relative error between the observed value of the sample point and the simulated analysis result.

The coefficients *b* = (*β*_1_, *β*_2_, … *β_n_*) of the basic functions can be determined by the least-squares method, as follows:(2)$$b={\left(\phi^{T}\phi \right)}^{-1}(\phi^{T}y)$$
where *b* = (*β*_1_, *β*_2_… *β_n_*) and matrix *ϕ* is
(3)$$\phi =\left[\begin{array}{ccc} \varphi_{1}{\left(x,y\right)}_{1} & \ldots & \varphi_{N}{\left(x,y\right)}_{1}\\ \ldots & \ldots & \ldots \\ \varphi_{1}{\left(x,y\right)}_{N} & \ldots & \varphi_{N}{\left(x,y\right)}_{N} \end{array}\right]$$
where *N* is the number of sample points. Coefficient *b* can be determined by substituting Equation (3) into Equation (2).

The accuracy of the surrogate models must be assessed using multiple criteria, namely, the coefficient of multiple determination (*R*^2^), relative error (*RE*), root mean square error (*RMSE*), and adjusted coefficient of multiple determination (*R*^2^*_adj_*), which are written as
(4)$$RE=\frac{\tilde{y_{i}}-y_{i}}{y_{i}},R^{2}=1-\frac{SSE}{SST},R_{\mathrm{adj}}^{2}=1-\frac{N-1}{N-n}(1-R^{2}),RMSE={\left(\frac{SSE}{N-n-1}\right)}^{0.5}$$
where *y_i_* is the simulation result, *SSE* is the total sum of the squares, and *SST* is the sum of squares of the following residuals:(5)$$SST=\sum_{i=1}^{N}{\left(y_{i}-\bar{y}\right)}^{2},SSE=\sum_{i=1}^{N}{\left(y_{i}-\tilde{y}\right)}^{2}$$
where $\bar{y}$ is the mean of *y_i_*.

## 4. Cold Roll-Forming Analysis of the Flat Strip

### 4.1. Design of Experiment

The roll gap *g_r_* and roll spacing *s_r_* are designated as design variables. The strip’s roll gap varies from 0.30 mm to 0.50 mm, and the roll spacing changes from 90 mm to 110 mm. The cold roll-forming process simulation of the strip is time-consuming, and establishing the surrogate models requires enough sample points. Hence, a two-factor five-level full-factorial experimental design is used to obtain 25 design points. Table 1 shows the step size and range of the design variables of the strip during the cold roll-forming procedure.

### 4.2. Finite Element Model

The upper and lower rollers are modeled as rigid bodies to minimize computational cost. The strip is made of 5052 aluminum alloy. The material’s properties are listed in Table 2, while the stress and strain during the plastic deformation stage are presented in Table 3. The stress–strain curve of the material is shown in Figure 5.

The strip was defined as an extruded shell consisting of 2613 nodes and 2400 four-node reduced integrated shell elements (S4R) with the enhanced hourglass control option enabled. The mesh size of the strip is configured to be 2 mm. This class of elements considers only the linear component of the nodal incremental displacement, thereby significantly reducing the computational time. The eight rollers were defined by 14,638 nodes, 14,419 linear quadrilateral elements of type R3D4, and 438 linear triangular elements of type R3D3. The finite element model and geometric dimensioning of the cold roll-forming procedure are shown in Figure 6.

### 4.3. Analysis Steps

A flower diagram of a strip with L and V cross-sections according to the principle of cubic curves of the horizontal projected trajectory of the end of the vertical edge is shown in Figure 7.

The roller surfaces of the eight rollers were connected to control reference points through rigid body constraints. The interactions between the different parts were constructed by self-contact, and normal behavior was set as hard contact. The contact characteristics of the interaction are, respectively, normal behavior and tangential behavior. The friction formula for tangential behavior is penalty contact, which is suitable for most metal-forming problems, with a friction coefficient of 0.3. The first pass is established between the first set of the upper rollers and the lower rollers. The cross-section is unchanged after the strip passes through the first pass. The quasi-static process of the strip cold roll-forming was analyzed using the ABAQUS/Explicit procedure. The minimization of inertia forces’ impact during simulation analysis necessitated the utilization of a smooth step loading method and the application of a specific viscous pressure on the surfaces of the strip. In the 6.5 s roll-forming process, the four upper rollers rotated simultaneously around the *y*-axis at a speed of 2.4 rad/s, while the four lower rollers rotated in sync around the same axis at a speed of −2.4 rad/s. The remaining five degrees of freedoms (DoFs) of the eight rollers were constrained. The DoF of the strip is fully constrained in the initial analysis step. The strip progresses along the z-axis at a velocity of 80 mm/s during the roll-forming analysis step.

The simulation results of the cold roll-forming process, which lasted for 6.5 s, are shown in Figure 8. The maximum stress and equivalent plastic strain of the strip are investigated under different roll gaps (0.3–0.5 mm) and four sets of roll distances (90–110 mm), taking into account the influence exerted by each set of rolls. The FEA results for the strip’s cold roll-forming process at the 25 design points are listed in Table 4.

### 4.4. Surrogate Models

The RS models of the maximum stress *S_r_*_-*max*_ and equivalent plastic strain *S_ep_* during the roll-forming process can be derived by Equations (4) and (5) after obtaining the sample points of the first roll-forming process based on the simulation results in Table 4. Therefore, the quadratic response polynomial functions of *S_r_*_-*max*_ and *S_ep_* are derived as
(6)$$\begin{array}{ll} s_{r-\max} & =20246.3864645-44248.6562087g_{r}-623.8688666s_{r}+144564.1549150g_{r}^{2}\\ & +221.6009531g_{r}s_{r}+8.7981610s_{r}^{2}-240171.027967\times g_{r}^{3}-78.9274589g_{r}^{2}s_{r}\\ & -2.0010392g_{r}s_{r}^{2}-0.0549071s_{r}^{3}+141113.2554194\times g_{r}^{4}+144.7262602g_{r}^{3}s_{r}\\ & -0.3442728g_{r}^{2}s_{r}^{2}+0.0075918745g_{r}s_{r}^{3}+0.0001266\times s_{r}^{4} \end{array}$$
(7)$$\begin{array}{ll} S_{ep}= & -177.5818154-562.721767g_{r}+9.962978882s_{r}+224.1947039g_{r}^{2}\\ & +13.72510054g_{r}s_{r}-0.182835582s_{r}^{2}+892.1076328g_{r}^{3}-13.52280972g_{r}^{2}s_{r}\\ & -0.075898733g_{r}s_{r}^{2}+0.001348782s_{r}^{3}-521.6891725g_{r}^{4}-2.037760233g_{r}^{3}s_{r}\\ & +0.080078424g_{r}^{2}s_{r}^{2}+0.0000144099g_{r}s_{r}^{3}-0.00000342995s_{r}^{4}; \end{array}$$

The total relative errors between the response surface model and FE analysis are derived from Equations (4)–(7) based on Section 4.3, and listed in Table 5. The accuracy of the different RS models for the strip during the roll-forming process is listed in Table 6.

The values of *R*^2^ and *R*^2^*_adj_* are in close proximity to 1, indicating a high level of correlation between the variables. Additionally, the *RMSE* is sufficiently small, allowing for an accurate assessment of the precision of the RS model. The response surfaces of *S_r_*_-*max*_ and *S_ep_*, which are plotted with the roll gap *g_r_* and roll space *s_r_* of the strip, are shown in Figure 9. The maximum stress shows a positive correlation with the roll space, while demonstrating a negative correlation with the roll gap. The pattern displayed by strain is in direct contrast.

### 4.5. Optimization Design of Cold Roll-Forming

The maximum stress is selected as the objective, while the equivalent plastic strain acts as the constraint to mitigate material damage during the cold roll-forming process. The roll gap and roll space are considered independent variables. The optimization problem for the cold roll-forming process can be modeled as
(8)$$\mathrm{opt}.\left\{\begin{array}{l} S_{r-max}(g_{r},s_{r})\leq 255\mathrm{Mpa}\\ \mathrm{Min}:S_{ep}(g_{r},s_{r})\\ 0.3\mathrm{mm}\leq g_{r}\leq 0.5\mathrm{mm}\\ 90\mathrm{mm}\leq s_{r}\leq 110\mathrm{mm} \end{array}\right.$$

The optimization process in ISIGHT 2020 software utilizes the Large Scale Generalized Reduced Gradient (LSGRG) algorithm. The maximum number of iterations is set to 40, while convergence epsilon is set to 40 and 0.001, respectively. The quantity of relative step sizes is equivalent to 0.001. The number of convergence iterations amounts to 3. The value of the binding constraint epsilon is equal to 1 × 10^−4^. The maximum number of failed runs is five, and both the penalty for failed run and the objective values are equal to 1 × 10^30^. The scale factors for *S_r_*_-*max*_ and *S_ep_* are both set to *S*_1_ = *S*_2_ = 1.0. The weight factors for *S_r_*_-*max*_ and *S_ep_* are set as *w*_1_ = 0.01 and *w*_2_ = 10, respectively.

The optimal design is obtained by solving the optimization model Equation (8) using the LSGRG algorithm through ISIGHT, with roll gap *g_r_* = 0.47 mm and roll space *s_r_* = 91.10 mm. The optimized roll gap and roll space obtained from the first optimization are utilized as the initial parameters for the second helical wrap-forming optimization process.

## 5. Helical Wrap-Forming Process

### 5.1. Finite Element Model of the Helical Wrap-Forming Process

Optimizing the cross-sectional shape of the strip in this step involves determining the L-edge height (*h_L_*) and the V-corner angle (*β_v_*), as illustrated in Figure 10.

The design variables include the L-edge height *h_L_* and the V-corner angle *β_v_*. The height of the L-edge varies from 2.8 mm to 3.2 mm, while the V-corner angle ranges from 35° to 45°. The configuration of the four sets of rollers and strips remains unchanged from that of the initial roll-forming process. The mandrel is designed as a rigid body. The angle formed between the strip and the mandrel is commonly referred to as the feed angle *α_f_*, which can be determined by the following equation:(9)$$\alpha_{f}=\arctan\left(B_{s}/\Pi d_{m}\right)$$
where *B_s_* represents the width of the strip cross-section after the roll-forming process, and *d_m_* is the mandrel diameter.

Equation (9) can be solved by substituting *B_s_* = 32 mm and *d_m_* = 36 mm, enabling us to calculate *α_f_* = 74.6°.

The structural distributing coupling links the midpoint of the strip helical wrapping end cross-section to its control point. The four upper rollers were displaced by 25 mm along the *x*-axis, while the four lower rollers underwent a displacement of −25 mm in the roll-forming process. The four upper rollers then rotated around the *y*-axis at a velocity of 2.4 rad/s, while the four lower rollers rotated in the opposite direction at a velocity of −2.4 rad/s. The mandrel was rotating at a speed of 6.28 rad/s around its axis. The roll-forming step lasted for 0.2 s, while the helical wrap-forming step was 4 s. The simulation results of the helical wrap-forming process are shown in Figure 11. The maximum stress *S_r_*_-*max*_ and wrapping torque *T_w_* during the helical wrap-forming procedure for 25 sample points are listed in Table 7.

### 5.2. Optimization Design of Helical Wrap-Forming

The RS models of the maximum stress *S_w_*_-*max*_ and wrapping torque *T_w_* during the helical wrapping process can be derived by Equations (4) and (5), based on the simulation results presented in Table 7. The quadratic response polynomial functions of *S_w_*_-*max*_ and *T_w_* are derived as
(10)$$\begin{array}{ll} s_{w-\max}= & -40930.842+71922.02h_{L}-1180.463\beta_{v}-42058.255h_{L}^{2}\\ & +824.69790h_{L}\beta_{v}+13.61064\beta_{v}^{2}+11609.103h_{L}^{3}-474.5222h_{L}^{2}\beta_{v}\\ & +14.44517059h_{L}\beta_{v}^{2}-0.578634\beta_{v}^{3}-1130.575h_{L}^{4}+44.61378h_{L}^{3}\beta_{v}\\ & +0.976745h_{L}^{2}\beta_{v}^{2}-0.167502h_{L}\beta_{v}^{3}+0.00664\beta_{v}^{4}; \end{array}$$
(11)$$\begin{array}{ll} T_{w}= & -13357.100-1346.931h_{L}+1541.110\beta_{v}+5276.90h_{L}^{2}\\ & -752.2851h_{L}\beta_{v}-31.25836289\beta_{v}^{2}-1705.1765h_{L}^{3}+132.7835h^{2}\beta_{v}\\ & +9.56464h_{L}\beta_{v}^{2}+0.290660716\beta_{v}^{3}+174.9829h_{L}^{4}-10.8185025h_{L}^{3}\beta_{v}\\ & -0.521458h_{L}^{2}\beta_{v}^{2}-0.055463h_{L}\beta_{v}^{3}-0.000793\beta_{v}^{4}; \end{array}$$

The total REs between the RS model and FE results are listed in Table 8. The coefficient of multiple determination *R*^2^ and adjusted *R*^2^_adj_ of *S_w_*_-*max*_ and *T_w_* are 0.9846, 0.9631, 0.9704, and 0.9290, respectively, which confirm the accuracy of the different RS models in predicting the helical wrapping process.

The helical wrapping tube will be utilized in the construction of a mechanism for space-related applications. The torque magnitude during the wrapping process is directly correlated, to a certain extent, with the stiffness of the helical wrapping tube fitting. Therefore, optimizing the maximum wrapping torque becomes an objective. The constraint of maximum stress during the strip wrapping process is considered in the optimization process outlined in Section 4.5.
(12)$$\mathrm{opt}.\left\{\begin{array}{l} \mathrm{Max}:T_{w}(h_{L},\beta_{v})\\ S_{w-\max}(h_{L},\beta_{v})\leq 280\mathrm{Mpa}\\ 2.8\mathrm{mm}\leq h_{L}\leq 3.2\mathrm{mm}\\ 35{}^{\circ}\leq \beta_{v}\leq 45{}^{\circ} \end{array}\right.$$

The Sequential Quadratic Programming (NLPQLP) algorithm is utilized for the optimization process. The termination accuracy, Rel step size, and Min Abs step size are set to 50, 1 × 10^−6^, 0.001, and 1 × 10^−4^, respectively. The weight factors of *S_w_*_-*max*_ and *T_w_* are set as *w*_1_ = 0.01 and *w*_2_ = 0.1.

After solving the optimization model Equation (12) using the NLPQLP algorithm, the optimal design is obtained, i.e., the height of the L-edge *h_L_* = 2.96 mm and the V-corner angle *β_v_* = 43.98°. The optimized roll gap and roll space obtained from the initial optimization will be utilized as the starting parameters for the second helical wrap-forming optimization.

### 5.3. Parameter Study

The influence curves of the L-side height and the V-angle on maximum stress *S_w-max_* and torque *T_w_* are depicted based on the sample point data provided in Table 7, as illustrated in Figure 12, which are established by linear regression. The torque in the strip wrapping process is observed to escalate proportionally with the augmentation of L-side height. The torque, on the other hand, diminishes as the V-angle increases. The stress in the strip wrapping process and its magnitude decrease with an increase in both the length of the L-side and the V-angle. The stiffness of the helical roll-forming tube can be enhanced, and material damage during the wrapping process can be prevented by appropriately increasing the height of the L-edge and reducing the V-angle.

## 6. Conclusions

This study introduces a compact and lightweight helical forming apparatus designed for the manipulation of metal tubes in orbital applications. The structure consists of four sets of rollers, the wrapping mandrel, the locking roller, and the flat strip storage component. The flat strip underwent a series of four rollers to achieve the L+V cross-section, and subsequent helical wrapping around the mandrel in order to form the securely locked metal tube. The tubes can be applied for in-orbit assembly of super-sized structures.

The rolling and helical wrapping process of the metal strip was optimized using a two-step approach. The FE simulation models for roll-forming and helical wrapping were separately established. The proposed approach presents a two-step optimization method to enhance the maximum stress, equivalent plastic strain, and wrapping torque of roll-forming L+V cross-sections and wrap-forming helical tubes. The optimized sample points in the second step are established based on the optimal roll distance and roll gap obtained in the first optimization step, namely, roll gap *g_r_* = 0.47 mm and roll space *s_r_* = 91.10 mm. The optimal L-side height and V-angle of the strip cross-section in the second step are determined to be *h_L_* = 2.96 mm and *β_v_* = 43.98 °, respectively.

The impact of certain roll-forming parameters on the maximum stress, equivalent plastic strain, and wrapping torque of both the cross-section and the rollers was investigated. In the roll-forming process, the maximum stress and equivalent plastic strain decrease as the roll gap *g_r_* and roll space *s_r_* increase; however, an excessively long roll gap can lead to a suboptimal cross-section forming effect, while an excessively large roll distance will require larger equipment. The wrapping torque in the strip wrapping process increases proportionally with the height of the L-side, while it decreases inversely with an increase in the V-angle. The maximum stress during the wrapping process decreases with an increase in both L-side length and V-angle. The stiffness of the helical tubes can be enhanced and material damage during the wrapping process can be prevented by increasing the L-edge height and reducing the V-angle.

## Figures and Tables

**Figure 1 materials-17-04832-f001:**
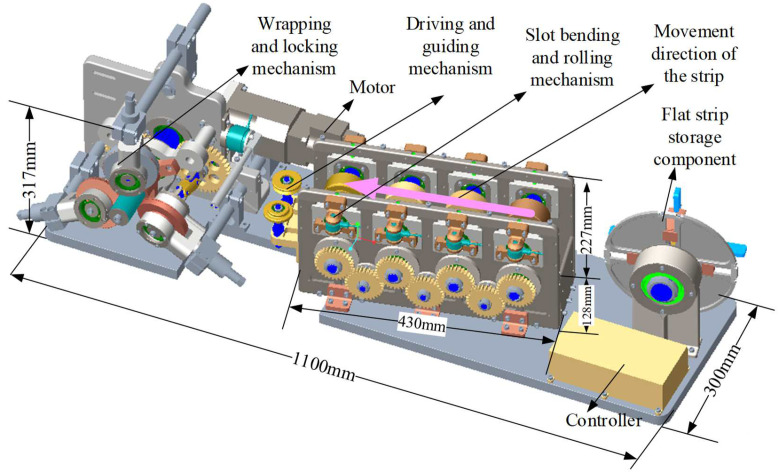
The three-dimensional configuration of the metal strip helical forming apparatus.

**Figure 2 materials-17-04832-f002:**
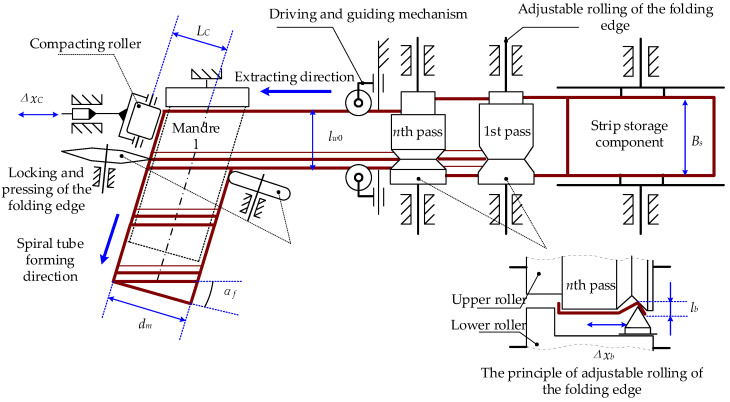
Schematic diagram of metal cold wrap- and roll-forming apparatus.

**Figure 3 materials-17-04832-f003:**
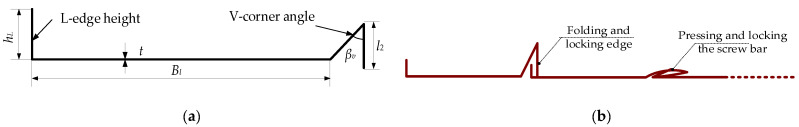
Strip cross-section configuration: (**a**) L+V cross-section, (**b**) strip wrapping and locking principle.

**Figure 4 materials-17-04832-f004:**
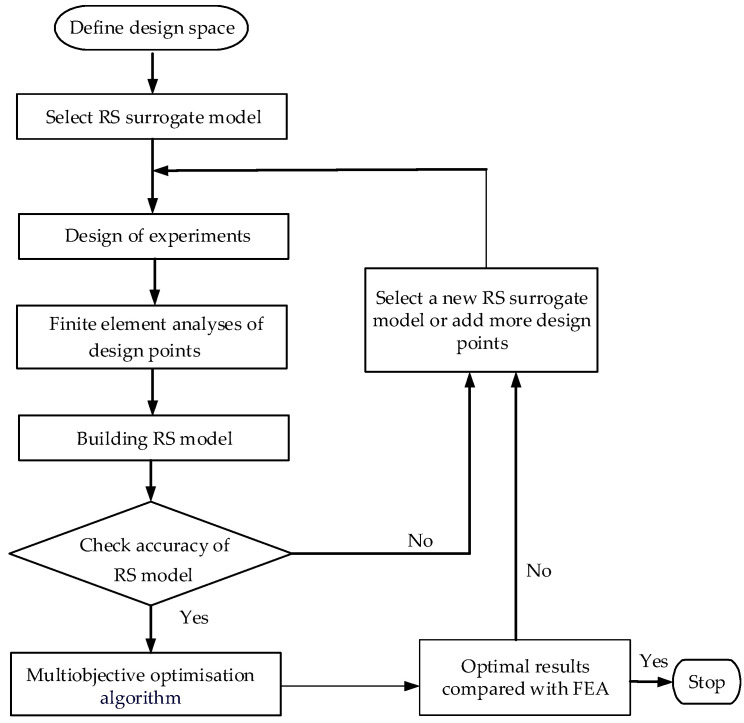
Flowchart of RSM optimization.

**Figure 5 materials-17-04832-f005:**
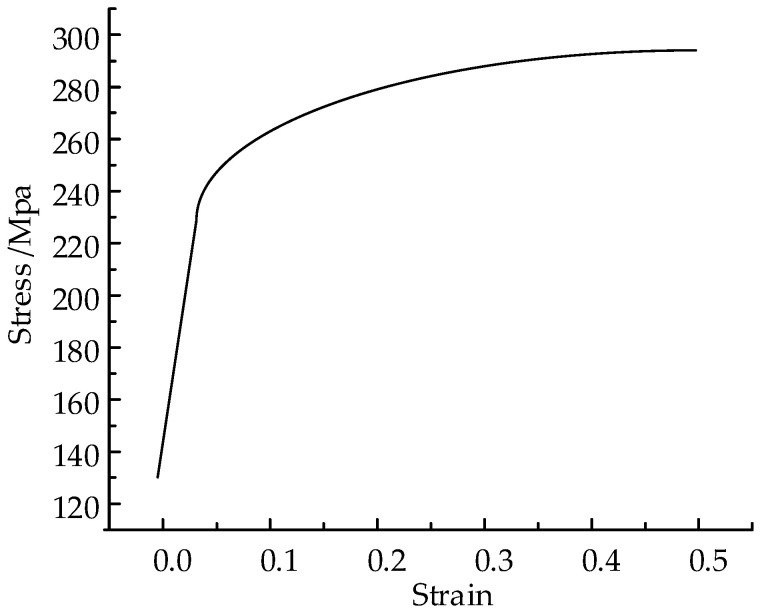
The stress–strain curve of the material.

**Figure 6 materials-17-04832-f006:**
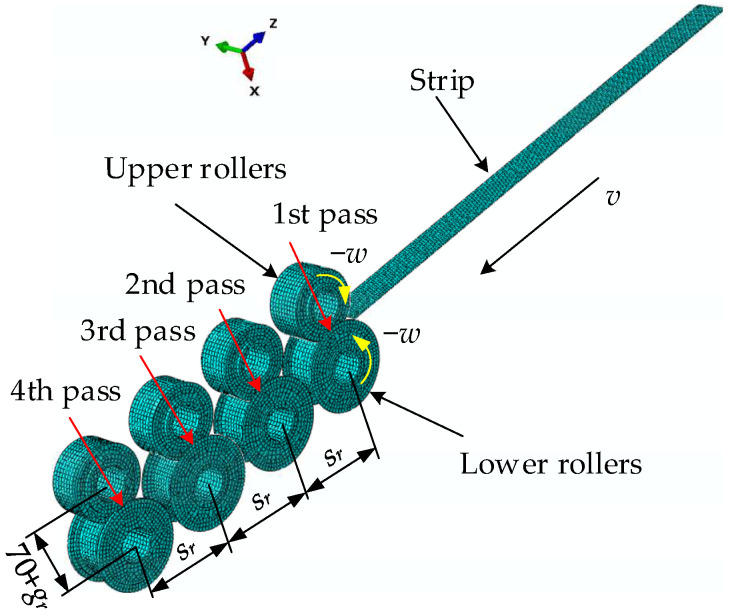
Finite element model and geometric dimensioning of the cold roll-forming procedure.

**Figure 7 materials-17-04832-f007:**
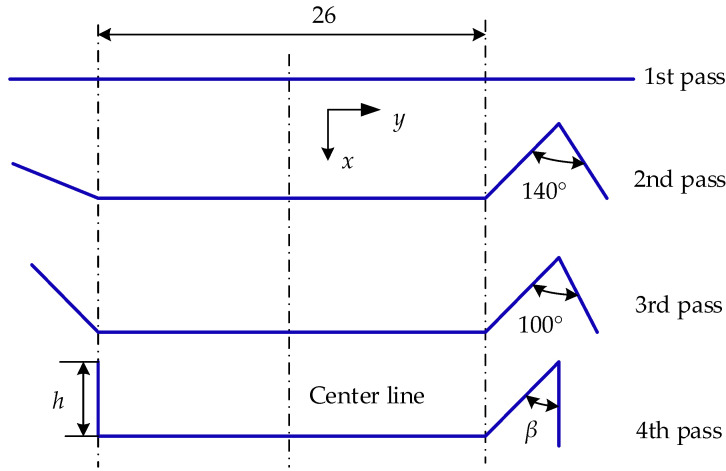
Flower diagram of a strip with L+V cross-section.

**Figure 8 materials-17-04832-f008:**
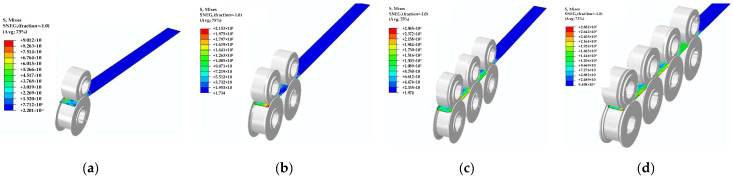
The stress program of the four-roller rolling process. (**a**) First pass; (**b**) second pass; (**c**) third pass; (**d**) fourth pass.

**Figure 9 materials-17-04832-f009:**
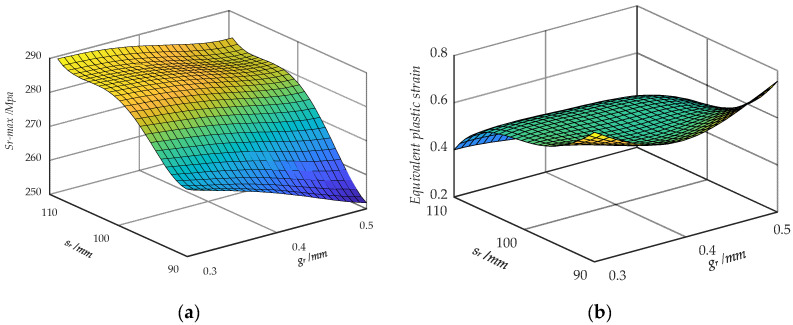
Response surface of *S_r_*_-*max*_ and *S_ep_*. (**a**) Response surface of *S_r_*_-*max*_; (**b**) response surface of *S_ep_*.

**Figure 10 materials-17-04832-f010:**
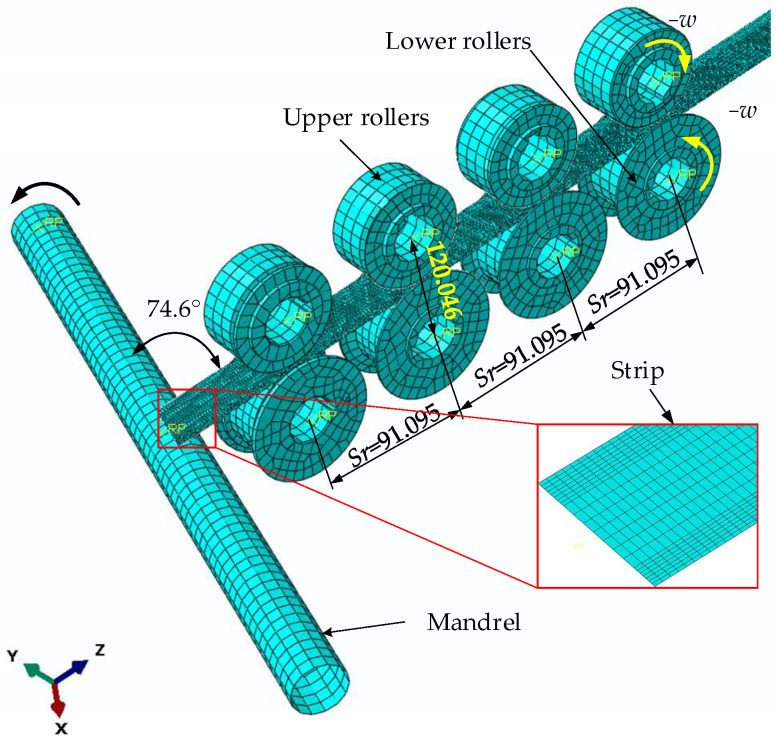
Finite element model and geometric dimensioning of the cold wrap-forming procedure.

**Figure 11 materials-17-04832-f011:**
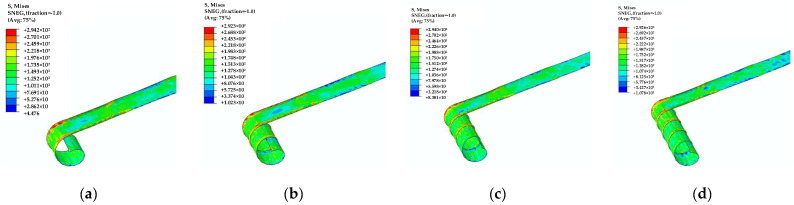
The stress image of four helical circle wrapping processes. (**a**) First circle; (**b**) second circle; (**c**) third circle; (**d**) fourth circle.

**Figure 12 materials-17-04832-f012:**
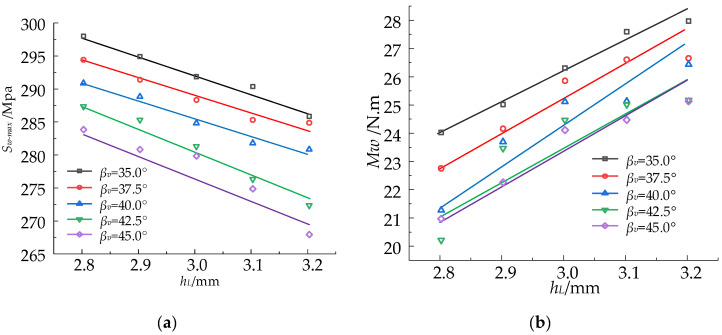
Variation in *S_w-max_* and *T_w_* with *h_L_* and *β_v_*. (**a**) Effect of *h_L_* on *S_w-max_*; (**b**) effect of *h_L_* on *T_w_*; (**c**) effect of *β_v_* on *S_w-max_*; (**d**) effect of *β_v_* on *T_w_*.

**Table 1 materials-17-04832-t001:** Step size and range of design variables of the strip during the cold roll-forming procedure.

Parameter	Step Size	Lower Bound	Upper Bound
Roll gap *g_r_*/mm	0.05	0.30	0.50
Roll spacing *s_r_*/mm	5	90	110

**Table 2 materials-17-04832-t002:** Material properties of 5052 aluminum alloy.

Density/(g/mm^3^)	Young Modulus/Mpa	Poisson’s Ratio
0.00270	70,300	0.33

**Table 3 materials-17-04832-t003:** The stress–strain parameters of 5052 aluminum alloy during the plastic deformation stage.

Stress/Mpa	132	172	201	228	246	262	277	291	296
Strain	0	0.0271	0.0688	0.135	0.198	0.257	0.271	0.454	0.502

**Table 4 materials-17-04832-t004:** Maximum stress *S_r_*_-*max*_ and equivalent plastic strain *S_ep_* during the first roll-forming process.

No.	Roll Gap *g_r_*/mm	Roll Spacing *s_r_*/mm	Maximum Stress *S_r_*_-*max*_/Mpa	Equivalent Plastic Strain *S_ep_*
1	0.30	90	283.52	0.76
2	0.30	95	284.77	0.64
3	0.30	100	288.84	0.61
4	0.30	105	291.05	0.66
5	0.30	110	290.56	0.76
6	0.35	90	280.52	0.61
7	0.35	95	282.25	0.59
8	0.35	100	283.74	0.59
9	0.35	105	285.52	0.58
10	0.35	110	288.28	0.56
11	0.40	90	278.01	0.58
12	0.40	95	279.67	0.58
13	0.40	100	281.39	0.55
14	0.40	105	284.38	0.54
15	0.40	110	286.02	0.54
16	0.45	90	264.35	0.55
17	0.45	95	268.29	0.55
18	0.45	100	271.91	0.54
19	0.45	105	274.63	0.54
20	0.45	110	274.08	0.48
21	0.50	90	252.34	0.40
22	0.50	95	255.74	0.39
23	0.50	100	261.84	0.37
24	0.50	105	265.02	0.35
25	0.50	110	270.31	0.30

**Table 5 materials-17-04832-t005:** *RE* errors between the FE and RS results for the strip during the roll-forming process.

No.	FE Results		RS Results		*RE* (%)	
	*S_r_*_-*max*_/MPa	*S_ep_*	*S_r_*_-*max*_/MPa	*S_ep_*	*S_r_* _-*max*_	*S_ep_*
1	283.52	0.76	283.64	0.75	0.04	−1.24
2	284.77	0.64	285.24	0.64	0.16	1.37
3	288.84	0.61	288.32	0.61	−0.18	1.01
4	291.05	0.66	290.46	0.66	−0.20	0.008
5	290.56	0.76	291.14	0.75	0.20	−0.71
6	280.52	0.61	280.18	0.62	−0.11	2.86
7	282.25	0.59	281.68	0.58	−0.20	−0.90
8	283.74	0.59	284.50	0.56	0.26	−4.63
9	285.52	0.58	286.51	0.57	0.34	−0.34
10	288.28	0.56	287.48	0.58	−0.27	3.07
11	278.01	0.58	277.93	0.57	−0.02	−1.06
12	279.67	0.58	279.55	0.57	−0.04	−0.86
13	281.39	0.55	282.26	0.56	0.31	2.89
14	284.38	0.54	284.21	0.55	−0.01	2.41
15	286.02	0.54	285.45	0.51	−0.20	−3.29
16	264.35	0.55	265.27	0.54	0.34	−0.84
17	268.29	0.55	267.79	0.55	−0.18	0.44
18	271.91	0.54	271.08	0.54	−0.30	1.58
19	274.63	0.54	273.57	0.53	−0.38	−1.63
20	274.08	0.48	275.59	0.48	0.55	0.55
21	252.34	0.40	251.76	0.39	−0.23	0.75
22	255.74	0.39	256.51	0.38	0.29	−0.21
23	261.84	0.37	261.61	0.36	−0.08	−0.85
24	265.02	0.35	265.79	0.34	0.29	−0.62
25	270.31	0.30	269.65	0.30	−0.24	1.08

**Table 6 materials-17-04832-t006:** Accuracy of the different RS models for the strip during the roll-forming process.

	*R* ^2^	*R* ^2^ _adj_	*RE*
*S_r_* _-*max*_	0.9957	0.9897	[−0.38% 0.55%]
*S_ep_*	0.9915	0.9796	[−4.63% 3.07%]

**Table 7 materials-17-04832-t007:** Maximum stress *S_r_*_-*max*_ and wrapping torque *T_w_* during helical wrap-forming procedure.

No.	L-Edge Height *h_L_*/mm	V-Corner Angle *β_v_*/°	Maximum Stress *S_w_*_-*max*_/Mpa	Wrapping Torque *T_w_*/Nm
1	2.8	35.0	297.96	24.02
2	2.8	37.5	294.40	22.74
3	2.8	40.0	290.85	21.27
4	2.8	42.5	287.32	20.20
5	2.8	45.0	283.82	20.95
6	2.9	35.0	294.90	25.01
7	2.9	37.5	291.37	24.15
8	2.9	40.0	287.80	23.68
9	2.9	42.5	284.29	23.46
10	2.9	45.0	280.80	22.25
11	3.0	35.0	291.85	26.30
12	3.0	37.5	288.30	25.85
13	3.0	40.0	284.78	25.11
14	3.0	42.5	281.29	24.46
15	3.0	45.0	277.82	24.10
16	3.1	35.0	290.32	27.58
17	3.1	37.5	285.29	26.59
18	3.1	40.0	281.79	25.12
19	3.1	42.5	278.31	25.01
20	3.1	45.0	274.85	24.46
21	3.2	35.0	285.82	27.96
22	3.2	37.5	274.85	26.65
23	3.2	40.0	278.81	26.43
24	3.2	42.5	275.35	25.17
25	3.2	45.0	271.91	21.12

**Table 8 materials-17-04832-t008:** RE errors between the FE and RS results for the strip during the helical wrapping process.

No.	*S_w_*_-*max*_/Mpa		*T_w_*/Nm		*RE* (%)	
	FE Results	RS Results	FE Results	RS Results	*S_r_* _-*max*_	*T_w_*
1	297.96	297.97	24.02	23.83	0.004	−0.75
2	294.40	294.77	22.74	22.86	0.12	0.52
3	290.85	291.45	21.27	21.36	0.20	0.48
4	287.32	286.40	20.20	20.50	−0.32	1.51
5	283.82	284.23	20.95	20.68	0.14	−1.28
6	294.90	294.84	25.01	25.27	−0.01	1.07
7	291.37	291.01	24.15	24.33	−0.12	0.77
8	287.80	288.52	23.68	23.32	0.25	−1.52
9	284.29	284.19	23.46	22.85	−0.03	−2.55
10	280.80	281.07	22.25	22.83	0.09	2.62
11	291.85	292.79	26.30	26.47	0.32	0.65
12	288.30	287.55	25.85	25.61	−0.25	−0.91
13	284.78	285.36	25.11	24.98	0.20	−0.48
14	281.29	281.46	24.46	24.69	0.06	0.98
15	277.82	277.34	24.10	24.12	−0.17	0.09
16	290.32	290.79	27.58	27.30	0.16	−1.01
17	285.29	284.04	26.59	26.40	−0.43	−0.71
18	281.79	282.28	25.12	25.91	0.17	3.15
19	278.31	279.20	25.01	25.42	0.32	1.68
20	274.85	274.69	24.46	23.79	−0.05	−2.70
21	285.82	285.12	27.96	28.08	−0.24	0.42
22	274.85	277.41	26.65	26.85	0.93	0.79
23	278.81	276.89	26.43	26.09	−0.68	−1.23
24	275.35	275.67	25.17	24.86	0.11	−1.18
25	271.91	272.07	21.12	21.50	0.06	1.79

## Data Availability

Data are contained within the article.

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
