# Peer review of "Optimization of Mechanical Properties during Cold Rolling and Wrap-Forming for Metal Helical Tubes"

_materials, 2024, doi:10.3390/ma17194832_

Round 1

Reviewer 1 Report

Comments and Suggestions for Authors

This is an interesting article where the authors propose a novel approach for a helical forming device for metal tubes. The investigation of the process through FEM analysis allows for the optimization of a two-step approach and the selection of process parameters that reduce equipment load while increasing the quality of the rolled strip and final product. The article is well-written, and the results are of both scientific and practical interest. The reviewer recommends the article for publication after addressing a few minor concerns:

- The stress-strain curve used in the FEM simulation should be shown, either in a plot or as an equation.

- The rounding of the physical parameters should be revised. For example, in the conclusion, the values in millimeters should be presented with two decimal places.

- The parameter study shown in Fig. 11, in its initial approximation, can be analyzed with the help of linear regression. This model can be very useful for engineers to understand how changes in parameters can impact the results.

Reviewer 2 Report

Comments and Suggestions for Authors

The paper is well written and addresses a problem of high industrial interest. Since advanced manipulations, like ones studied in the paper, are difficult to be verified with experiments, authors could use simpler shapes to compare. In addition using of artificial intelligence is gaining importance for optimal design of processes like this. Authors can find relevant material in recent publications like: Spathopoulos SC, Stavroulakis GE. Springback Prediction in Sheet Metal Forming, Based on Finite Element Analysis and Artificial Neural Network Approach. Applied Mechanics. 2020; 1(2):97-110. https://doi.org/10.3390/applmech1020007

Reviewer 3 Report

Comments and Suggestions for Authors

The mention of mechanical properties in the title of the article gives a misleading impression of its content. In this respect, the manuscript neither contains the problem nor the methods used to solve it. With regard to the results presented, there is a lack of information on the parameters of the equipment and on the experimental verification of the results.

Reviewer 4 Report

Comments and Suggestions for Authors

The article concerns interesting metal-forming topics, which have potentially interesting industrial uses. The paper presents a very intriguing concept of combining the open profile roll-forming process with bending the pipe afterwards. Advanced numerical modelling was used, and the analysis of selected indicators/optimal parameters for such configured technology was presented. Both numerical models and the analytical description of the issue were discussed very accurately. The concept of the article as a theoretical consideration looks fine. However, there is a significant lack of experimental verification, at least in selected parts of the simulation results. Moreover, a few detailed remarks/questions are addressed to the authors, which are issues to clarify or develop.

The article has been well prepared, and the photos are good quality. The proposed conclusions summarize the adequately. The literature survey is fair. Some language errors indicate that the text has yet to be subjected to critical language analysis, causing a clear understanding of the authors' thoughts. It can be essential in the case of non-English authors because, very often, the effect of such neglect is a misunderstanding by the reader/reviewer of the appropriate, substantive content of the article. Suggestions for improvement have been issued to the authors.

GENERAL REMARKS

1. The authors define the rollers as rigid areas and then devote the number of nodes and elements used to represent them. In most FEM software, the lines/surfaces describe rigid (not deformable) objects. Among others, the simplification relies on this procedure. However, some software uses finite elements to define rigid entities. Is it a case for ABAQUS/Explicit?

2. The model is expensive from the point of view of the calculation time, as the authors recall, but has any optimisation of the model been carried out (number/size of elements)? Or was it based on previous work in this area?

3. How was the friction phenomenon modelled, and what friction coefficient was utilised?

DETAILED REMARKS

1. Figure 1. The three-dimensional configuration of metal strip helical forming.

For a cursory look at the figure, it is not clear, in what sense was the "three-dimensional" term used, since the drawing is not spatial and do show the equipment/station from one direction, from above? However, the picture presents partially 2D and partially 3D visualisation for a closer look, isn't it? Consider changing the figure for more clear visualisation.

2. These four sets of rollers extrude the two edges of the flat strip out of the lock groove.

What exactly the rollers do with the strip material? It is suggested to use the term "extrusion" very carefully. Extrusion is a term which is used strictly for specific processes in bulk metal forming. It occurs when the material is pressed against a die, passes through it and changes initial cross-section. From the mechanics point of view, the stress state - three dimensional compression - in formed material placed in front of the die shaping zone is one of the basic features. The rollers can DIRECT or MOVE strip, etc., but not extrude the edges of the strip.

3. The eight roller was defined by 14638 nodes, 14419 liner quadrilateral elements of type R3D4,...

It is not clear: the eight rollers or the eight roller? Every roller has 14638 nodes, etc...? or all rollers collects these numebers for nodes and elements recalled in text?

Comments on the Quality of English Language

The level of English used can be essential in the case of non-English authors. Very often may cause a misunderstanding by the reader/reviewer of the appropriate, substantive content of the article. A few detailed examples are taken to show suggestions.

General remark

Some sentences require correction from the point of view of either language elegance or correctness. I leave it to the authors' decisions and possibly the editor. Sentence constructions such as the following suggest the need to use native speaker or at least quite good available tools of automatic language correction. Please consider this possibility.

Detailed remarks

Introduction

1. The space truss, an integral component of spacecraft structures, is primarily comprised of a specific number of one-dimensional rods ...

The phrase "is primarily comprised of" may be wordy. Consider changing to:

The space truss, an integral component of spacecraft structures, comprises a specific number of ...

2. It has found extensive applications in the construction of deep space ...

Same situation, wordy phrase. Suggestion:

It has found extensive applications in constructing deep space ...

3. The future exploration of space will necessitate the utilization of larger, lightweight, high-performance, and cost-effective space trusses, such as in-orbit service platforms.

This sentence may be unclear or hard to follow. Consider rewritning:

Future space exploration will necessitate utilising larger, lightweight, high-performance, and cost-effective space trusses, such as in-orbit service platforms.

4. The space truss structure utilized for in-orbit operation in current spacecraft typically ranges from 10 to 100 meters [1-3]. Future space exploration will necessitate utilising larger, lightweight, high-performance, and cost-effective space trusses, such as in-orbit service platforms.

Authors use both North American and British spelling (utilized, utilising,...). Both are acceptable, but it's best to be consistent. The same applies to behavior/behaviour, modeling/modelling, etc.

5. ...has the potential to enable on-orbit wrap of metal tube fittings ...

It seems that there is an article usage problem. Consider:

...has the potential to enable THE on-orbit wrap of metal tube fittings ...

6. ...thereby enhancing the overall rigidity of tube fittings ...

Again wordy phrase. Consider:

...enhancing the rigidity of tube fittings ...

7. In the process of forming tube fittings...

It appears, that "in the process of" is unnecessarry in this sentence. Consider remowing it:

In forming tube fittings ...

8. In forming tube fittings, the bending of metal strip induces elastoplastic deformation.

The sentence is repeated twice.

9. Theoretical methods were utilized to solve the displacement, stress, and strain of thin-walled sheets under pure bending conditions [8].

Consider rewriting the sentence for better clarity:

Theoretical methods were utilized to solve thin-walled sheets' displacement, stress, and strain under pure bending conditions [8].

10. Liu [15] found that the deformation behavior of double wall brazed tube  ...

Liu [15] found that the deformation behavior of THE double wall brazed tube ...

11. Murugesan [18] investigated the presence of longitudinal bow, the reason behind flange height deviation, spring-back, and identification ...

Murugesan [18] investigated the presence of A longitudinal bow, the reason behind flange height deviation, spring-back, and THE identification ...

12. Tajik [19] studied the twist defect by finite element (FE) analysis for asymmetrical channels with different flange lengths and found the twist defect  ...

Tajik [19] studied the twist defect by finite element (FE) analysis for asymmetrical channels with different flange lengths and found THAT the twist defect ...

13. Weiss [20] studied the roll forming of two coils exhibiting coil sets and found that there were significant differences ...

It seems that "that there were" maybe unnecessary. Consired removing this phrase:

Weiss [20] studied the roll forming of two coils exhibiting coil sets and found significant differences ...

14. The process of optimizing the flat strip roll forming V+L section is carried out, resulting in the determination of the optimal roll gap and roll space.

Optimizing the flat strip roll forming V+L section is carried out, determining the optimal roll gap and roll space.

15. The second step involves studying the optimized parameters of the sample points determined by the experimental of design (DoE) ...

The second step involves studying the optimized parameters of the sample points determined by the experimental design (DoE) ...

or

The second step involves studying the optimized parameters of the sample points determined by the design of experiment (DoE) ...

16. The strip will pass through the flat strip storage component, ...

Maybe it is better to use Simple Present tense, since authors describe a general concept which works already.

The strip passes through the flat strip storage component, ...

17. During the wrap- and roll- forming process, the strip is transferred from the strip storage component to the lock slot bending device and sequentially passes through four roller groups.

Sounds better:

During the wrap- and roll- forming process, the strip is transferred from the strip storage component to the lock slot bending device and sequentially passes through four groups of rollers.

Round 2

Reviewer 4 Report

Comments and Suggestions for Authors

The authors referred to all comments.